# Convolutional neural network-based classification of glaucoma using optic radiation tissue properties
John Kruper [1,2], Adam Richie-Halford [3], Noah C. Benson [2], Sendy Caffarra[3,4], Julia Owen[5,6], Yue Wu[5,6], Catherine Egan [7], Aaron Y. Lee[5,6], Cecilia S. Lee [5,6], Jason D. Yeatman[3], Ariel Rokem [1,2] ✉ & UK Biobank Eye and Vision Consortium*

## Abstract

**Background** Sensory changes due to aging or disease can impact brain tissue. This study aims to investigate the link between glaucoma, a leading cause of blindness, and alterations in brain connections.
**Methods** We analyzed diffusion MRI measurements of white matter tissue in a large group, consisting of 905 glaucoma patients (aged 49-80) and 5292 healthy individuals (aged 45-80) from the UK Biobank. Confounds due to group differences were mitigated by matching a sub-sample of controls to glaucoma subjects. We compared classification of glaucoma using convolutional neural networks (CNNs) focusing on the optic radiations, which are the primary visual connection to the cortex, against those analyzing non-visual brain connections. As a control, we evaluated the performance of regularized linear regression models.
**Results** We showed that CNNs using information from the optic radiations exhibited higher accuracy in classifying subjects with glaucoma when contrasted with CNNs relying on information from non-visual brain connections. Regularized linear regression models were also tested, and showed significantly weaker classification performance. Additionally, the CNN was unable to generalize to the classification of age-group or of age-related macular degeneration.
**Conclusions** Our findings indicate a distinct and potentially non-linear signature of glaucoma in the tissue properties of optic radiations. This study enhances our understanding of how glaucoma affects brain tissue and opens avenues for further research into how diseases that affect sensory input may also affect brain aging.

## Plain Language Summary

In this study, we explored the relationship between glaucoma, the most common cause of blindness, and changes within the brain. We used data from diffusion MRI, a measurement method which assesses the properties of brain connections. We examined 905 individuals with glaucoma alongside 5292 healthy people. We refined the test cohort to be closely matched in age, sex, ethnicity, and socioeconomic backgrounds. The use of deep learning neural networks allowed accurate detection of glaucoma by focusing on the tissue properties of the optic radiations, a major brain pathway that transmits visual information, rather than other brain pathways used for comparison. Our work provides additional evidence that brain connections may age differently based on varying sensory inputs.

Glaucoma is the leading cause of irreversible blindness[1]. The disease causes retinal ganglion cell (RGC) death, and consequently disconnection of the transmission of visual information through the optic nerve to the lateral geniculate nucleus (LGN). The optic radiations (OR) are the brain white matter connections that further transmit the information from the LGN to the visual cortex. Though the cells in the LGN whose axons constitute the OR are not directly affected by glaucoma, they are deprived of their sensory input. A major question in sensory neuroscience, with significant clinical implications, is whether such a change to the sensory periphery affects the properties of central processing pathways[2,3]. Examining the properties of the OR in glaucoma provides an opportunity to study the downstream effects of changes to the sensory periphery on central brain connections. Alternatively, one of the hypotheses about the effects of glaucoma on the white matter is that it represents accelerated aging, at least within the retina[4].

---

[1]Department of Psychology, University of Washington, Seattle, WA, USA. [2]eScience Institute, University of Washington, Seattle, WA, USA. [3]Graduate School of Education and Division of Developmental Behavioral Pediatrics, Stanford University, Stanford, CA, USA. [4]University of Modena and Reggio Emilia, Modena, Italy. [5]Department of Ophthalmology, University of Washington, Seattle, WA, USA. [6]Roger and Angie Karalis Johnson Retina Center, Seattle, WA, USA. [7]Moorfields Eye Hospital, NHS Trust, London, UK. *A list of authors and their affiliations appears at the end of the paper. ✉e-mail: arokem@uw.edu

Diffusion MRI (dMRI), which measures the random motion of water within brain tissue[5], is a non-invasive method to reconstruct the trajectory of white matter pathways, such as the OR, and to assess the physical properties of the tissue within them. DMRI has previously been used to measure the properties of white matter tissue in subjects with glaucoma, with studies showing a range of different, and sometimes contradicting, effects[6–11]. In several of these studies, changes were not specific to the visual pathways[12–15], suggesting wide-spread reorganization or systemic changes related to glaucoma that also manifest in white matter changes.

Automated Fiber Quantification[16] (AFQ) is an automated method used to quantify tissue properties from dMRI data. In multi-shell dMRI measurements, tissue properties can be quantified using the diffusional kurtosis imaging model (DKI)[17,18]. Statistics derived from this model are sensitive to biological changes, such as aging and disease, and, when they are used in concert, can help constrain the interpretations of the underlying biological processes[18–21]. AFQ calculates one-dimensional tract profiles of white matter tissue properties dervied from DKI, such as mean diffusivity (MD), which measures average molecular motion of water molecules and is sensitive to tissue density; fractional anisotropy (FA), which measures the directional selectivity of water molecule diffusion and is sensitive to myelination[19], but also to axon fiber coherence and crossing of multiple fibers within a voxel; and mean kurtosis (MK), which measures the non-Gaussianity of water diffusion and is sensitive to tissue complexity[17].

A powerful paradigm for detecting differences between groups—e.g., subjects with glaucoma and a control group—is to create machine learning (ML) systems that classify these groups in a held-out dataset. ML algorithms, and particularly convolutional neural networks (CNNs)[22], can capitalize on high-dimensional and/or non-linear patterns in data, such as the complex configuration of tissue properties along the length of a brain tract, to discriminate between different categories of individuals. They can be more accurate than other models, overcoming significant variation across individuals, but they also require large amounts of data. The UK Biobank (UKBB) dataset is the largest dMRI dataset to date with many thousands of subjects[23]. The large sample size of the UKBB provides an unprecedented opportunity to study nuanced and potentially non-linear correspondence between glaucoma and white matter tissue properties with ML methods. The large dataset can also be used to mitigate confounding effects of variables that are correlated with a disease state (for example, glaucoma is more prevalent in older individuals) using statistical matching[24].

In the present study, we use data from the UKBB to fit CNNs that classify individuals with glaucoma based on the tissue properties of their OR, and two other control tracts. We delineate these tracts automatically, in each individual, using an open-source software that we have developed[25], which implements AFQ in the Python programming language (https://yeatmanlab.github.io/pyAFQ). To mitigate confounds because of biases in the characteristics of subjects with glaucoma, we create a sub-sample of healthy subjects with closely matched age, sex, ethnicity, and socio-economic background. We find that the CNNs trained on the ORs exhibit higher accuracy in classifying subjects with glaucoma when contrasted with CNNs trained on control tracts. To test the accelerated aging hypothesis about the effects of glaucoma on the white matter, we also assess generalization of the CNN trained to classify glaucoma in classifying subjects from different age groups. In addition, we test whether the effects of glaucoma on OR tissue properties generalize to other retinal diseases, by assessing whether the CNN trained to classify glaucoma could also classify UKBB subjects with age-related macular degeneration (AMD). We find that CNNs do not generalize across classification tasks. Taken together, these results suggest potentially non-linear and glaucoma-specific differences in OR tissue properties between subjects with and without glaucoma, that do not resemble accelerated aging.

## Methods
### Data access
Data was obtained through the UK Biobank health research program[23]. De-identified Magnetic Resonance Imaging (MRI) and health data were downloaded from the UK Biobank repository and our study, which did not involve human subjects research, was exempt from IRB approval. The UK Biobank study itself was approved by an IRB under 11/NW/0382. Informed consent was obtained from all UK Biobank participants during an initial two and a half hour recruitment visit, which included broad consent for use of their anonymized samples in a range of different health-related research (see[23] for details). The UK Biobank's research ethics committee approval means that researchers wishing to use the resource do not need separate ethics approval, unless re-contact with participants is required (which is irrelevant in our case). Analysis of de-identified human data is not considered human subject research by the University of Washington Institutional Review Board and does not require additional approval and an exemption was determined.

### Datasets and statistical matching
From the available data, we created several distinct datasets, using statistical matching to create datasets where bias due to age and other factors is negligible. To be included in any of these, study participants must have had a dMRI data acquisition and a final visual acuity logMAR (log of the Minimum Angle of Resolution) of less than or equal to 0.3 if measured (from UKBB data field 5201). We re-ran all analyses without this exclusion criterion, and found the same results.

Dataset A was composed of the following sub-samples of the UKBB data:

1. Glaucoma sub-sample: we first selected 905 subjects classified as having glaucoma (in at least one eye) by the UK Biobank's Assessment Center Environment (ACE) touchscreen question: "Which eye(s) are affected by glaucoma" (see UKBB data field 6119).
2. Control sub-sample: We selected 5292 UKBB subjects for a control pool[23,26,27]. These subjects answered "no eye problems/disorders" to the ACE touchscreen question: "Has a doctor told you that you have any of the following problems with your eyes? (You can select more than one answer)" (see UKBB data field 6148).

To reduce bias from age and other potential confounders, we used statistical matching[24] to create a "matched dataset". We calculated the Mahalanobis distance[28] between the confounders of all pairs of glaucoma and control subjects. The confounders[29] we used were age, sex, ethnicity, and the Townsend deprivation index (TDI)[30]. We used the 'linear_sum_assignment' method from Scipy 1.8.0[31] to match test and control subjects such that the total Mahalanobis distance between them is minimized. This is a modified implementation of the Jonker-Volgenant algorithm[32]. We then thresholded the matched dataset, only keeping matched subjects with a Mahalanobis distance of 0.3 or less.

In the full sample, glaucoma subjects tend to be older ($\mu \pm \sigma$ for glaucoma: 68 ± 7; control: 62 ± 7). After Mahalanobis distance matching[28], we created the matched dataset A with 856 glaucoma subjects and 856 control subjects of similar ages (Supplementary Fig. S1b; glaucoma: 68 ± 6, control: 68 ± 6). While Supplementary Fig. S1 only shows age matching, we simultaneously matched on sex, ethnicity, and TDI[30], with similar results (Supplementary Fig. S2).

After matching was concluded, we divided the resulting dataset— *dataset A*—into two groups: train (80%) and test (20%). From the train set, 20% was set aside as a validation set for hyperparameter selection. So, in total, the dataset is apportioned as follows: 64% for the train set, 20% for the test set, and 16% for the validation set. Matched control and glaucoma subjects were assigned into these groups in tandem. All decisions about model architecture and hyperparameters were made using dataset A training and validation sets. The results shown in this paper are from test sets, which were not viewed until final determinations about model architecture and hyperparameters were made.

To additionally test generalization of the glaucoma model, we constructed two additional *test datasets*. These datasets used control subjects that were not included in the glaucoma-matched pool ($N = 4456$):

1. AMD dataset (***dataset A.1***): Subjects were marked as having AMD based on the ACE touchscreen question in UKBB data (field 6148) described above. A set of matched controls was selected using statistical matching on the same criteria as above. This created a pool of 81 participants, out of which 78 matched pairs were found.

2. Aging dataset (***dataset A.2***): 70 year-old subjects were selected from the control pool and were matched to other subjects in the control pool, with the matching modified to match to subjects 10 years younger, in addition to the other criteria used above. This dataset had 166 exactly 70-year old subjects and 166 approximately 60-year old subjects.

Another dataset - ***dataset B*** - was sub-sampled from the UKBB to train age-group classification models. Here, we selected subjects with ages 70–79 ($N = 962$) from the control sub-sample. These were matched to other controls with the matching modified to match to subjects 10 years younger, in addition to the other criteria used above, to create 819 matched pairs. This dataset was also divided into train, test, and cross-validation the same way as dataset A.

To test the generalization from age-group classification models, we constructed two additional *test datasets*. Controls were taken from a control pool where participants selected for the dataset used to train the age-group classification models were removed ($N = 4473$):

1. Glaucoma (***dataset B.1***). In this dataset, only participants aged between 60-64 with glaucoma were included. This created a pool of 147 participants, out of which 142 matched pairs were found.

2. AMD (***dataset B.2***). Given lower prevalence of AMD in the overall dataset, we did not use an age inclusion criterion. This created a pool of 81 participants, out of which 80 matched pairs were found.

Though 27.6% of the subjects used in dataset A are in dataset B, there was no overlap between the training data and the test data in either of these datasets.

## MRI acquisition

We used preprocessed dMRI provided through the UKBB. The acquisition protocol is already described elsewhere[27]. Briefly, dMRI measurements were conducted with a spatial resolution of 2-by-2-by-2 $mm^3$, TE/TR = 14.92/3600 *msec*. With anterior-to-posterior phase encoding, there are five volumes with no diffusion weighting (b = 0), and 50 volumes each with b=1000 $s/mm^2$ and b=2000 $s/mm^2$. An additional 6 b=0 volumes were acquired with posterior-to-anterior phase encoding direction for EPI distortion correction. Preprocessing is also described elsewhere[26]. Briefly, the FSL "eddy" software was used to correct for head motion and eddy currents, including correction of outlier slices. Gradient distortion correction was performed and non-linear registration using FNIRT was calculated. The FNIRT mapping was also used to map the individual subjects data to the MNI template.

## Automated fiber quantification

AFQ[16] is an analysis pipeline that automatically delineates major white matter pathways and quantifies the properties of white matter tissue along the length of these major pathways. We used a Python-based open-source implementation of this pipeline (pyAFQ; https://github.com/yeatmanlab/pyAFQ)[25] to extract the tissue properties of OR sub-bundles and two control bundles - the corticospinal tract (CST) and the uncinate fasciculus (UNC) - that are not a part of the visual system. Using AFQ, white matter pathways were identified from a candidate set of streamlines based on anatomical landmarks: inclusion, exclusion, and endpoint regions of interest (ROIs) that were based on the known trajectory of the pathways (e.g., OR). These ROIs were transformed into each subject's diffusion MRI coordinate frame using the FNIRT non-linear warp provided by UKBB. For UNC and CST, we also used a population-based probabilistic atlas[33]. We seeded a GPU-accelerated residual bootstrap tractography algorithm[34] with 64 seeds uniformly distributed in the voxels of the inclusion ROIs to generate candidate streamlines which may follow the trajectory of the OR in each subject's data.

These candidate streamlines were filtered to recognize the OR with these conditions: streamlines (1) do not pass through the sagittal midline of the brain; (2) have at least one point that is within 3*mm* of both of the inclusion ROIs; (3) do not have any point that is within 3*mm* from the exclusion ROI; (4) terminate within 3*mm* of the two endpoint ROIs (one in the thalamus and the other in V1)[35,36]. They were further filtered using the standard AFQ cleaning to remove outliers in length and trajectory[16]. Control bundles were generated using the same pipeline as the OR, using inclusion ROIs[16] and excluding streamlines that cross the midline.

Tissue properties were calculated using the diffusional kurtosis model implemented in DIPY[18,37]. The tract profiles were computed at 100 points along each bundle, however we excluded the first and last 10 nodes, because their measured properties are more strongly influenced by partial volume effects with the gray matter. Further visualizations and analyses with machine learning used these one dimensional tract profiles.

## Machine learning

We trained 1D convolutional residual neural networks on the tract profiles from the optic radiation and two control bundles (CST, UNC) to predict whether a subject has glaucoma. The network architecture is identical to the one in Fawaz et al.[38]. Briefly, the network consisted of three residual blocks[39], each containing a series of 1D convolutional layers with kernel sizes of 8, 5, and 3, interspersed with batch normalization and ReLU activation layers. Within each block, there was a residual connection. The residual connection involved connecting the input of a residual block to the input of its subsequent layer using an addition operation. The number of filters in each block was 64, then 128, then 128 filters. The network is implemented as part of an open-source software package for analysis of tractometry data that we develop (https://yeatmanlab.github.io/AFQ-Insight)[40].

We trained one network per bundle for a total of three networks. The middle 80 nodes out of the 100 from the tract profiles were used. Thus the input to these networks consisted of 1D tract profiles of length 80 and with six channels, one for each tissue property (FA, MD, and MK) and each hemisphere (left and right). Tissue properties were normalized by converting to Z-scores before use in the neural network. In subjects where some bundles were not found, missing bundles were imputed using the mean profile of that bundle for each tissue property, separately in the train and test data[41].

Independently, we trained 3 L2-regularized logistic regression models on the bundles. We used the liblinear solver in Scikit Learn 1.0.0 to fit these models[42,43]. To reduce overfitting seen in the validation set, we only used every other node from the 20th to 78th node out of 100 nodes before testing. We also used the validation set to determine the level of regularization.

We applied the networks trained on dataset A to datasets A.1 and A.2. Finally, we trained three more neural networks of the same architecture on the same bundles, using dataset B. We also tested its generalizability on its corresponding B.1 and B.2 datasets.

The results of the CNNs and logistic regression are compared using areas under curve (AUC) from a receiver operating characteristic (ROC) curve[44]. Depending on the curve, we asked one of two questions: (1) For the visual system tests (datasets A and B), are the OR AUCs significantly different from the control AUCs, and (2) for the visual system tests and the generalization tests (datasets A.1, A.2, B.1, B.2), are any of the AUCs significantly above chance. For question 1, we compared the OR AUC to each control bundle. We calculated a p-value from the DeLong test[44] and then corrected for the two comparisons using the False Discovery Rate (FDR) p-value correction[45]. We called the AUCs significantly different if the corrected p-value is below 0.05. For question 2, we used the variance calculated by the DeLong formula[44] to make a 95% confidence interval. We applied the Bonferroni correction to the confidence intervals[46] to correct for our testing of three different bundles. If the corrected confidence intervals did not include an AUC of 0.5, we called the AUC significant. The more conservative Bonferroni correction is used to correct for multiple comparisons in this case, because it is applied to confidence intervals, rather than to p-values.

## Reporting summary

Further information on research design is available in the Nature Portfolio Reporting Summary linked to this article.

## Results

In dataset A (glaucoma subjects statistically matched to controls, see "Methods" for more details), the glaucoma subjects have lower MK than the control subjects in their optic radiations across the bundle. They have lower FA and higher MD in the posterior OR, but slightly higher FA and lower MD in the anterior OR. This is shown in the Left OR column of Fig. 1, where the 95% confidence intervals between the orange and blue lines do not overlap. With this large sample size, even the small differences in the mean shown here are statistically significant. However, while the mean tract profiles differ significantly, the underlying distributions are highly overlapping, as shown by the thin interquartile range lines. Additionally, there are significantly different mean tissue properties in the uncinate (UNC) and corticospinal tract (CST). For example, there is a statistically significant difference in MK at position 35 in the left CST, and in FA at position 40 in the left UNC. In summary, differences in mean tract profiles are small (if statistically significant) and are not specific to the visual system. The same results are found in the right hemisphere (Supplementary Fig. S3).

Despite the high overlap between the distributions, CNNs trained on the tissue properties of white matter bundles in dataset A can classify participants with glaucoma and controls with an area under curve (AUC) that is greater than chance (Fig. 2). However, the only AUC that shows a statistically significant difference from chance is from the OR tissue properties

(AUC = 0.69). We also find that the CNN trained on OR tissue properties has a significantly higher AUC than a CNN trained on CST (DeLong's test: p = 0.0028 ) and a CNN trained on UNC (p = 0.0002). This indicates that differences in tissue properties between participants with glaucoma and controls are specific to the OR, and much weaker or non-existant in the non-visual control bundles.

Not all bundles were found in each subject. We calculated that, for dataset A, the extant bundle percentage for each bundle was: left UNC 100.0%, right UNC 100.0%, left CST 99.8%, right CST 99.8%, left OR 95.8%, right OR 94.9% (Supplementary Fig. S4). Note that the OR are found less often than the control bundles. This is because its high curvature makes it difficult to track. To test whether the results depend on these differences in missing data, we created equivalent ROC curves using only test subjects where all bundles are found, which shows similar effects (Supplementary Fig. S5).

As another control, we used a model of substantially less complexity than a CNN. We trained three logistic ridge regression models (Fig. 3). We used all three tissue properties (FA, MD, MK) from 30 positions along the core white matter from each hemisphere, for a total of 180 features (the choice of these features is elaborated upon in the Machine Learning section of the Methods). If regression models can predict glaucoma with an accuracy comparable to CNNs, this would indicate that the relationship between glaucoma and the tissue properties can be captured by a linear model and does not require the use of a CNN. Again, only the AUC from the model trained on the OR (0.63) is significantly different from chance. The OR AUC is significantly different from the UNC AUC (p = 0.0431), but just barely,

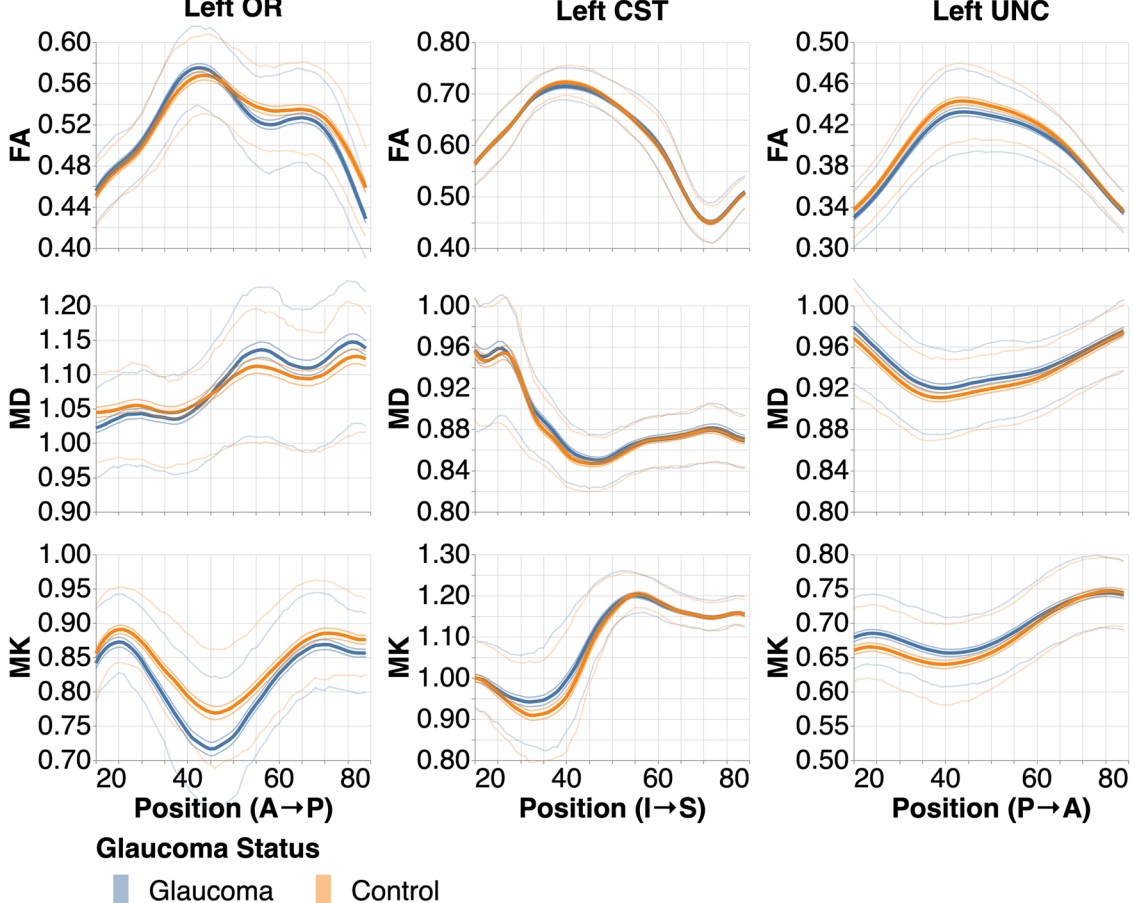

**Fig. 1 | Tract profiles in individuals with glaucoma and healthy controls.** Thick lines show the mean tract profiles in the left hemisphere of all bundle and tissue property combinations. The medium-thickness lines hugging the thick lines show the 95% confidence interval. The thin lines show interquartile ranges (*n* = 895 in each group). Positions in OR are from anterior to posterior (A → P), in the corticospinal tract (CST) are from inferior to superior (I → S), and in the uncinate (UNC) from posterior to anterior (P → A).

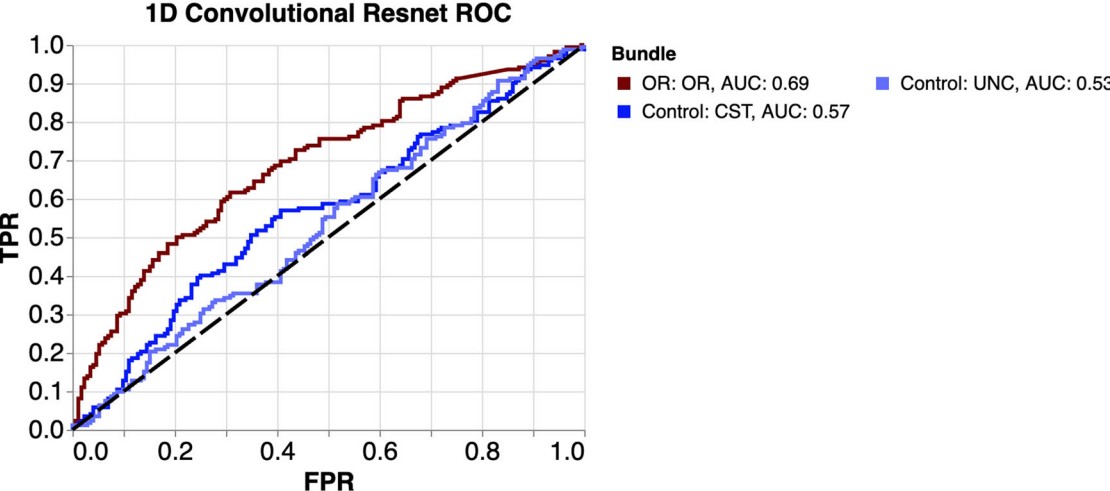

**Fig. 2 | Receiver operating characteristic (ROC) curves for prediction of glaucoma using the three neural networks.** Each were trained using three tissue properties from both hemispheres, but each from a different bundle. ROC curves compare true positive rate (TPR) to false positive rate (FPR) at various thresholds. Area under curve (AUC) is the summary of accuracy across all classification thresholds. Therefore, ROC/AUC metrics are invariant to the choice of decision criterion. Note that the OR in dark red have higher AUC than the control bundles.

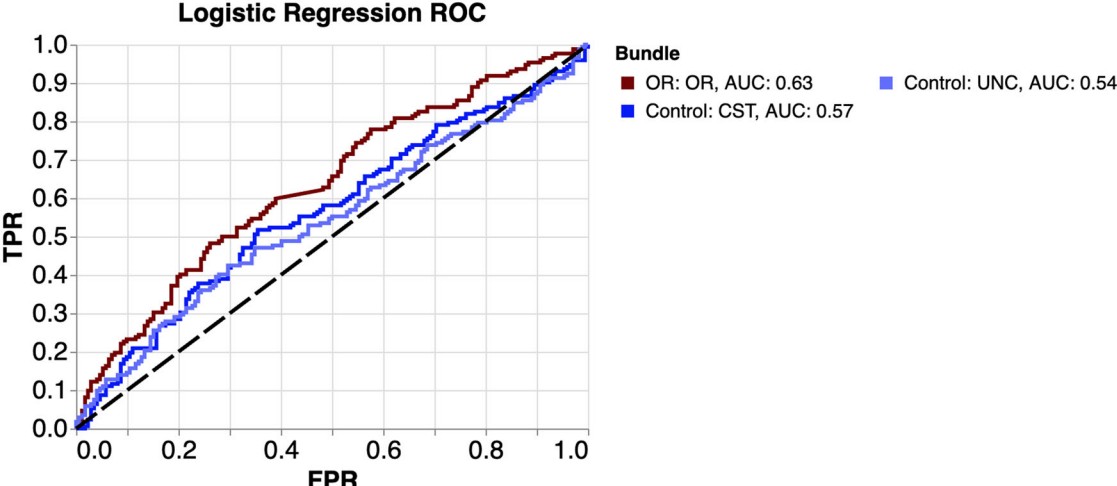

**Fig. 3 | ROC curves for the prediction of glaucoma using logistic regression.** Curves are colored by the bundle used for the classification. The OR AUC is lower when using logistic regression rather than the convolutional neural network (see Fig. 2 for comparison). As a result, the control bundles and OR have more similar AUC.

and it is not significantly difference from the CST AUC. Additionally, we found that the OR AUC from the CNN (Fig. 2), is significantly higher than the OR AUC from the logistic regression, though only marginally (DeLong's test: p = 0.0303). For more context on this marginal difference, we have provided the confusion matrices for these two models in Supplementary Table S1. The higher AUC of the OR CNN could be because the CNN better captures the complexity of the relationship between glaucoma and OR tissue properties. However, it is important to note that both of these AUC values are considered weak[47].

To test the generalization of our main results, we applied the glaucoma-trained network to datasets labeled according to AMD and age (datasets A.1, A.2, see "Methods"). We found no significant AUCs (no generalization; Fig. 4). Additionally, we trained three CNNs (one per bundle; the same bundles as used previously) on dataset B (70–79 year olds, with matched controls 10 years younger, see Methods) to classify age-group (70 vs. 60 year old). The AUCs for this network in the age-classification task are all between 0.61 and 0.64 (OR: 0.63, CST: 0.61, UNC: 0.64). However, none of the bundles have AUC significantly different from each other (p>0.05 for all comparisons using the DeLong's test), demonstrating that, unlike glaucoma, the tissue properties of the UNC and CST are just as informative for age

prediction as the tissue properties of the OR. Finally, we used the age-classification network to predict glaucoma and AMD (datasets B.1, B.2, see "Methods"). Here we found no significant AUCs in the glaucoma set (AUC, OR: 0.50; CST: 0.50; UNC: 0.53), nor in the AMD set (AUC, OR: 0.43; CST: 0.43; UNC: 0.42).

## Discussion

To study the effects of glaucoma on tissue properties in the white matter of the optic radiations, we used a large dataset of diffusion MRI measurements of participants with glaucoma from the UK Biobank[23], together with combinatorial methods to generate tightly-matched samples, automated methods for delineation of white matter tracts, and machine learning techniques. Previous research on glaucoma has included samples with a large diversity of characteristics[6–11]. We sub-sampled from the UK Biobank dataset to create a group of controls that closely matched 856 participants with glaucoma in age, sex, and socioeconomic status, focusing only on these 856 pairs of participants for our analysis. This approach allowed us to mitigate potential sampling biases that could confound the conclusions about how common the effects of glaucoma in the white matter are. Even after some participants are excluded to provide for this tight matching, the

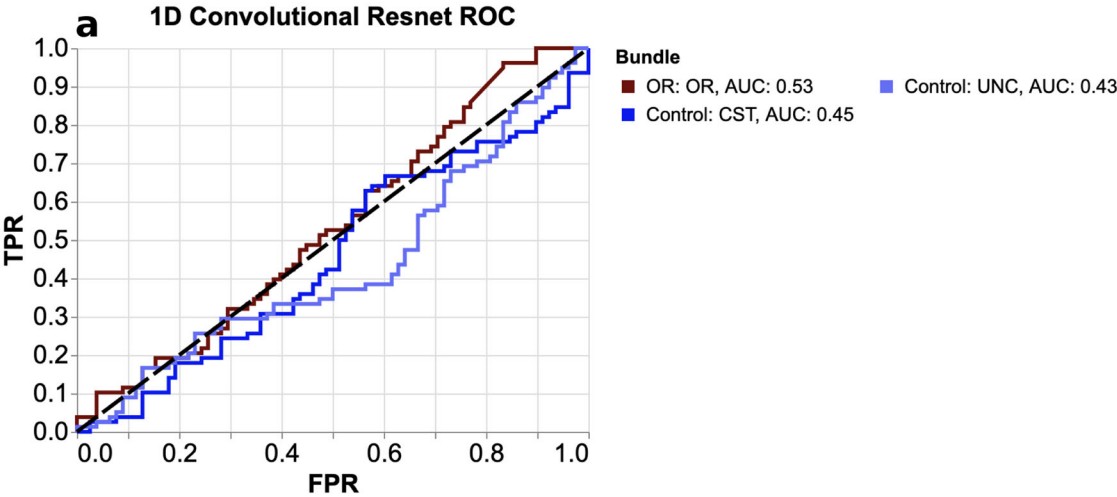

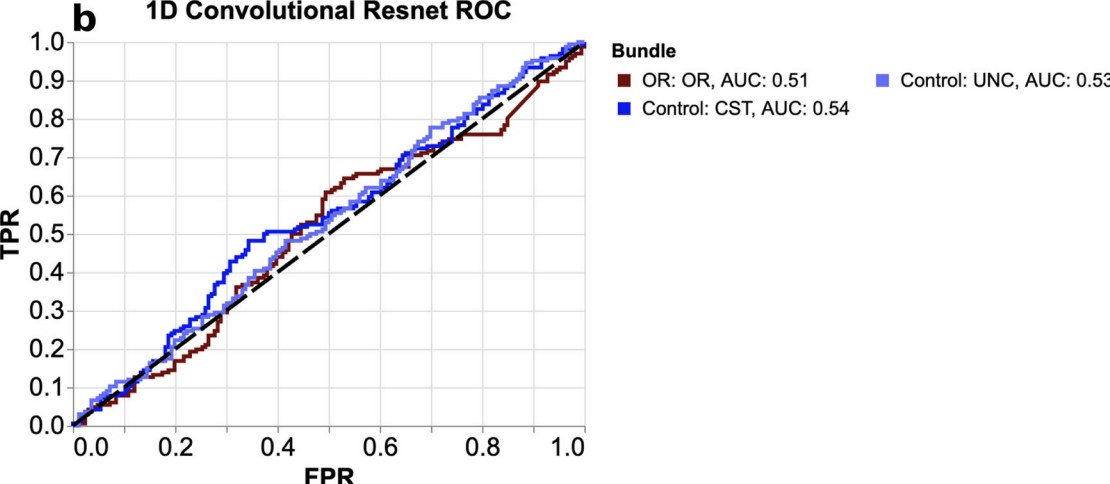

**Fig. 4 | Generarlization to other tasks.** ROC curves for prediction of AMD (**a**) and age (**b**) using the CNN trained on the glaucoma dataset. Classification performance did not significantly differ from chance for any of the networks. This demonstrates the inability of a network trained to distinguish subjects based on glaucoma status to generalize to age or AMD.

sample is approximately an order of magnitude larger than that of any previous study using dMRI in glaucoma. We found only small differences in tissue properties in the optic radiations, as well as in non-visual control tracts, with large overlaps in the distributions between participants with glaucoma and the tightly-matched control group. However, using convolutional neural networks, we found that classification of glaucoma participants is possible using the optic radiation with an AUC of 0.69. This is significantly higher than the AUCs of the non-visual control bundles, which themselves were not significantly different from chance. This provides evidence of specific effects of glaucoma detected in the visual white matter.

To test whether the effects of glaucoma on the white matter of the OR represent accelerated aging[4], We used the glaucoma-classification CNN to differentiate between subjects with a 10-year age gap (but matched on sex and socioeconomic status). We chose the ages 60 and 70, based on our previous observations that this is an age difference at which OR tissue properties change substantially[48], and also the age at which glaucoma risk increases substantially in the UKBB sample[49]. Nevertheless, the CNN trained to classify glaucoma did no better than chance in classifying subjects from different age groups. Furthermore, a CNN trained to perform the age classification task could not distinguish participants with glaucoma from their matched controls better than chance. This suggests that the features of the OR that are correlated with glaucoma status are different from those that

are correlated with the normal process of aging. In previous work with a much larger sample of UKBB subjects ages 45-81 ($n$ = 5382) we found that healthy aging is associated with substantial decreases in FA and MK and increases in MD[48]. In contrast, the lower values of FA, MK and MD that are indicative of glaucoma are consistent with more complex tissue configurations without substantial demyelenation or loss of tissue density[18]. Nevertheless, diffusion-derived tissue properties are sensitive to a variety of confounds. For example, FA is sensitive to demyelination[19], but also to the presence of crossing fibers[50], so it needs to be interpreted with caution.

We used a linear model—regularized logistic regression—and it was able to classify glaucoma with an AUC of 0.63. This AUC is significantly smaller than the AUC found when using the OR CNN to classify glaucoma (0.69). However, the $p$ value ($p$ = 0.0303) is proximate to the conventional threshold of 0.05, indicating marginal significance. The practical implications of this marginal increase in AUC are limited. ML algorithms have come under some criticism in their use in biomedical application, due to the "black box" nature of their operations, which can sometimes make them inscrutable[51]. Even if the OR CNN exhibited better performance in glaucoma classification, its results are less interpretable. In comparison, although the the logistic regression model would be more interpretable, it has a lower AUC for OR. The small significant difference between deep learning and linear models is also consistent with our recent findings regarding

differences between linear models and deep learning networks in modeling brain white matter development in a large sample of children and adolescents[52].

Differences between the OR of participants with glaucoma and healthy controls could arise for many reasons. One hypothesis is that the altered visual input due to the disease causes reorganization of the tissue responsible for downstream processing steps[53–55]. These downstream changes could also be related to transsynaptic degeneration through the LGN that can be measured through changes to the OR tissue properties. Indeed, previous research found that participants with glaucoma have significantly reduced LGN volumes[56]. While altered visual input may be a cause of these differences, we also showed that the glaucoma-trained CNN does not generalize to another retinal disease that causes altered visual input, age-related macular degeneration (AMD). The prevalence of AMD is much lower, and there is not enough data in the UKBB dataset to train a CNN using statistically matched AMD subjects, so we cannot rule out the possibility that an AMD-trained CNN would generalize to glaucoma. Moreover, the AMD sample is mostly composed of mild cases of AMD (with visual acuity that is close to intact), and it is possible that generalization would increase in a sample with more profound cases of AMD-related visual loss. Another possibility is that the ML algorithms may be specifically focusing on a subgroup of individuals with glaucoma who also have changes in the optic radiations and/or the LGN that lead to a retrograde optic nerve degeneration that is ascribed to glaucoma. In this case, the change to visual inputs is not the main driver of changes in the white matter.

Another set of hypotheses not addressed in the present study could relate glaucoma to Alzheimer's Disease (AD). AD and glaucoma are linked through their high levels of co-occurence, as well as through biochemical and pathophysiological similarities, and overlapping genetic mechanisms[57]. Therefore, it is an intriguing possibility that a glaucoma-trained CNN would generalize to AD, and vice versa. Unfortunately, the UKBB does not yet have a large enough sample of participants with dMRI measurement and a diagnosis of AD to test this question. At the same time, it is possible that comorbidities such as undiagnosed AD could be responsible for some of the differences between the glaucoma and control subjects. To further understand the relationship between glaucoma and AD, future work should apply the techniques described in this paper to other large datasets where these diseases are more prevalent, or to combine multiple, large datasets.

Importantly, the present study is purely correlational, and though we have accounted for some of the obvious confounding factors (such as age), we may have not accounted for all of the differences between participants with glaucoma and controls. For example, in previous work, we found that cardiovascular fitness variables are important features in classifying participants with glaucoma in the UK Biobank dataset[49]. In this dataset, we tried matching on cardiovascular variables but found that they did not change our results (not shown). Still, the effects measured in the OR and delineated with the CNN analysis could still reflect common underlying causes rather than direct effects of glaucoma on the tissue properties of the OR. Future studies could focus on other risk factors, such as a history of smoking, diabetes, high blood pressure, elevated intraocular pressure, or genetics, which might explain these differences.

Another limitation of the present study is that glaucoma and AMD status were determined based on self-report. This raises concerns that some of the individuals identified as having glaucoma have another eye disease, or no eye disease (false positives) and that there may be individuals with glaucoma among the controls (false negatives). One way in which false positives could affect our results is by the inclusion of people with other sources of sight impairments in the glaucoma group, and some of these could be impairments with a source directly in the OR. However, in our previous work with the UK Biobank dataset, we verified that self-report is highly congruent with ICD-10 diagnostic codes of glaucoma and that there was high repeatability of self-report among participants who reported glaucoma[49]. Furthermore, a previous study that examined glaucoma in the UK Biobank dataset[58] verified that the distributional properties of self-reported glaucoma match the distributional properties of glaucoma subjects

in several other population studies[59–61]. Even if one assumes that glaucoma self-report is rather accurate, within the glaucoma group there will be variations in disease sub-type (i.e., open-angle, normal pressure, etc.), along with variations in disease severity, duration and received treatments. Such a generalization could mask the differences in presentation across different types and stages of the disease. Future research to develop algorithms that can accurately classify the stage and type of the disease, providing even more fine-grained information about the link between retinal disease and brain tissue properties, will require such information. The presence of additional common eye disorders (e.g., high myopia, cataract) in the sample will also add variance that cannot be accounted for. The same would hold for the healthy controls, i.e., those who did no report the presence of glaucoma, but who well could suffer from other unreported diseases that could affect the retina and the visual pathways (e.g., multiple sclerosis, diabetes). These issues all highlight the limitations of the prospective approach used here.

Finally, there are limitations associated with the MRI methodology used. Due to the tract's high curvature, narrow sections, and intersections with other tracts, some OR were not detected in some subjects. Additionally, dMRI is unable to differentiate feedforward and feedback projections within the OR, so our conclusions encompass both.

Future work could improve each step of our analysis pipeline. In the present study we only considered the OR and not the rest of the retinogeniculate visual pathway. This is because the optic radiations are larger and easier to track than the smaller optic tract and optic nerves. A useful extension of this work would be to develop an algorithm which can recognize the optic tract reliably, automatically, and quickly, so that it can be deployed on thousands of subjects[62]. We used DKI[17,18] to model the tissue properties within the OR. This model augments the diffusion tensor imaging (DTI) model, removing the simplifying assumption of Gaussian diffusion. But, both of these models are phenomenological[63,64] and other models may provide a more mechanistic view of tissue properties that differ in participants with glaucoma. For example, the neurite orientation dispersion and density imaging (NODDI)[65] model, which explains the signal as the combination of different biophysically-interpretable tissue compartments. Future studies with UK Biobank and with other datasets could expand on our findings using NODDI and other modeling techniques. Once bundles were recognized and tract profiles were derived, we used a 1-D convolutional residual neural network to analyze their tissue properties. However, there are a variety of other neural network architectures that could be used[52]. Another relevant framework that could be applied to case-control studies is a normative modeling approach[66]. Normative modeling that uses deep learning neural networks has already previously been applied to tract profile data[67]. Normative modeling approches may also better handle the potential heterogeneity of glaucoma's effects on the white matter. Integrating more clinical information is another way to handle heterogeneity. The UK Biobank contains a plethora of clinical information that may be useful to consider as potential inclusion or exclusion criteria, or as confounders for statistical matching. For example optical coherence tomography (OCT) scans were collected in some subjects in the UK Biobank study. OCT scans could be compared with glaucoma status and white matter tissue properties as a mediator. It has already been shown that deep learning can be applied to OCT data to predict glaucoma[49] and even age-related macular degeneration[68].

In summary, we used neural networks to determine that there is a complex relationship between the tissue properties of the optic radiations and glaucoma. These differences persist even when participants with glaucoma are closely matched to controls. We did not find this relationship in non-visual control bundles. This relationship is not reflective of an accelerated aging process, but may instead reflect a change in the visual input and subsequent reorganization of the visual system, but it does not seem to generalize to other retinal disorders (AMD). Our study contributes to a growing body of research that utilizes CNNs to enhance biomedical imaging techniques. Importantly, the application of CNNs allows for the detection of subtle and non-linear effects that may otherwise go unnoticed. Although the present work is a novel application of deep learning in the field

of diffusion MRI, such techniques have already been used successfully in other fields, such as ophthalmology. In particular, there has been considerable success using retinal fundus images and machine learning to classify glaucoma[49,69]. Results from diffusion MRI serve to complement the previous findings by showing that brain and eye health are intricately linked. As the techniques used here are further developed, they may hold promise for future clinical applications[70], for example, as neural networks are used to establish baseline or normative models for different disease states[71].

## Data availability
This study uses publicly available data from the UK Biobank. More information on the data and access can be found here: https://www.ukbiobank.ac.uk/enable-your-research. Where possible, we have also shared aggregate data and trained models are available at https://github.com/36000/glauc_paper_scripts[72]. However, some figures are generated directly from individual particpants' data. This data cannot be made publicly available and must be accessed through the UK Biobank. All other data are available from the corresponding author on reasonable request.

## Code availability
All code to reproduce the analysis and the figures is available at https://github.com/36000/glauc_paper_scripts[72].

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

## Acknowledgements

This project was funded by NSF grant 1934292 (PI: Balazinska), NIH grant R01 AG 060942 (PI: Lee), NEI grant R01 EY033628 (PI: Benson), NIH grant RF1 MH121868 (PI: Rokem), NIH grant R01HD095861 (PI: Yeatman). J.K. was supported through the NSF Graduate Research Fellowship DGE-2140004. S.C. was funded by the "Rita Levi Montalcini" program, granted by the Italian Ministry of University and Research (MUR). Unrestricted and career development award from RPB (Julia Owen, Yue Wu, Cecilia Lee, Aaron Lee), Latham Vision Science Awards (Julia Owen, Yue Wu, Cecilia Lee, Aaron Lee), NEI/NIH K23EY029246 (Aaron Lee) and NIA/NIH U19AG066567 (Julia Owen, Yue Wu, Cecilia Lee, Aaron Lee, Ariel Rokem).

## Author contributions

J.K., A.R., C.S.L., J.O., A.Y.L., S.C., N.B., and J.Y. helped conceptualize the research ideas. J.K. and A.R. developed and implemented the software used. A.R.-H., S.C., and N.B. created models for the analysis. C.E., A.Y.L., C.S.L., and Y.W. organized and provided the data. A.Y.L. and C.S.L. provided the computing resources. J.K. and A.R. analyzed and investigated the data. J.K. and A.R. wrote the initial draft of the paper with input from all authors. All authors provided feedback on the manuscript.

## Competing interests

The authors declare the following competing interests: A.Y.L. reports grants from Santen, personal fees from Genentech, personal fees from US FDA, personal fees from Johnson and Johnson, grants from Carl Zeiss Meditec, personal fees from Topcon, personal fees from Gyroscope, non-financial support from Microsoft, grants from Regeneron, outside the submitted work;

This article does not reflect the views of the US FDA. All other authors declare that they have no conflicts of interest.

## Additional information

## UK Biobank Eye and Vision Consortium

Naomi Allen[8], Tariq Aslam[9], Denize Atan[10], Konstantinos Balaskas[7], Sarah Barman[11], Jenny Barrett[12], Paul Bishop[9], Graeme Black[9], Tasanee Braithwaite[13], Roxana Carare[14], Usha Chakravarthy[15], Michelle Chan[7], Sharon Chua[16], Alexander Day[7], Parul Desai[7], Bal Dhillon[17], Andrew Dick[10], Alexander Doney[18], Catherine Egan[7], Sarah Ennis[14], Paul Foster[16], Marcus Fruttiger[16], John Gallacher[8], David Garway-Heath[16], Jane Gibson[14], Jeremy Guggenheim[19], Chris Hammond[20], Alison Hardcastle[16], Simon Harding[21], Ruth Hogg[15], Pirro Hysi[20], Pearse Keane[16], Peng Tee Khaw[16], Anthony Khawaja[7], Gerassimos Lascaratos[7], Thomas Littlejohns[8], Andrew Lotery[14], Robert Luben[16], Phil Luthert[16], Tom MacGillivray[17], Sarah Mackie[12], Savita Madhusudhan[22], Bernadette McGuinness[15], Gareth McKay[15], Martin McKibbin[23], Tony Moore[16], James Morgan[19], Eoin O'Sullivan[24], Richard Oram[25], Chris Owen[26], Praveen Patel[7], Euan Paterson[15], Tunde Peto[15], Axel Petzold[27], Nikolas Pontikos[16], Jugnoo Rahi[28], Alicja Rudnicka[26], Naveed Sattar[29], Jay Self[14], Panagiotis Sergouniotis[9], Sobha Sivaprasad[7], David Steel[30], Irene Stratton[31], Nicholas Strouthidis[7], Cathie Sudlow[17], Zihan Sun[16], Robyn Tapp[26], Dhanes Thomas[7], Mervyn Thomas[32], Emanuele Trucco[18], Adnan Tufail[7], Ananth Viswanathan[7], Veronique Vitart[17], Mike Weedon[25], Katie Williams[20], Cathy Williams[10], Jayne Woodside[15], Max Yates[33] & Yalin Zheng[21]

[8]University of Oxford, Oxford, UK. [9]The University of Manchester, Manchester, UK. [10]University of Bristol, Bristol, UK. [11]University of Kingston, London, UK. [12]University of Leeds, Leeds, UK. [13]St Thomas' Hospital, London, UK. [14]University of Southampton, Southhampton, UK. [15]Queen's University Belfast, Belfast, UK. [16]UCL Institute of Ophthalmology, London, UK. [17]University of Edinburgh, Edinburgh, UK. [18]University of Dundee, Dundee, UK. [19]Cardiff University, Cardiff, UK. [20]King's College London, London, UK. [21]University of Liverpool, Liverpool, UK. [22]Royal Liverpool University Hospital, Liverpool, UK. [23]Leeds Teaching Hospitals NHS Trust, Leeds, UK. [24]King's College Hospital, London, UK. [25]University of Exeter, Exeter, UK. [26]St George's, University of London, London, UK. [27]UCL Institute of Neurology, London, UK. [28]UCL Institute of Child Health, London, UK. [29]University of Glasgow, Glasgow, UK. [30]Newcastle University, Newcastle, UK. [31]Gloucestershire Hospitals NHS Foundation Trust, Gloucester, UK. [32]University of Leicester, Leicester, UK. [33]University of East Anglia, Norwich, UK.

