## [Peer Review File · Communications Medicine]

Reviewers' comments:

Reviewer #1 (Remarks to the Author):

The authors present a study on the effect of glaucoma on the brain. They take a novel approach by training ML algorithms to classify the data into glaucoma patients and controls. Their main finding is that whereas a CNN can learn to do this, a conventional linear regression approach cannot. Moreover, they find that the ability to do this classification is specific to the OR and fails on other, non-visual, tracts, as well as on classifying age-defined cohorts. This suggests that glaucoma has a unique non-linear influence on the integrity of the OR.

This manuscript is interesting for a few reasons: first, by using data from the UKBiobank, the study analyzes a very large sample size of glaucoma patients. Secondly, the authors use a thorough approach to find closely matching control participants for the glaucoma participants. Thirdly, the authors compare their OR results to those of two non-visual tracts. Fourth, they train a series of CNNs and do some interesting cross comparisons to determine how specific the glaucoma-related findings are.

The manuscript/study also has some weaknesses. Foremost, the separation of the OR into various bundles results in a rather unexpected finding. Namely that the foveal and macular bundles, rather than the peripheral one, allows the classification. This is quite counterintuitive, as glaucoma typically affects the peripheral visual field. Rather than that this unexpected finding results in the authors questioning their approach, they come up with an, in my view, somewhat flawed explanation for it in terms of functional remapping from the peripheral visual field to the center. Secondly, they do not put much effort into attempting to explain what the "unique non-linear influence" might constitute, in particular at the biological level. Thirdly, the present manuscript is not well written and structured, but this could presumably be remedied.

Major issues

The authors separate the OR into three bundles (fovea, macular, peripheral). Surprisingly, the foveal and macular OR bundles enable the best classification even though glaucoma is known to foremost affect the peripheral visual field. The authors come up with an explanation for this in terms of plasticity, but I believe their reasoning on this is flawed. In my reasoning, if indeed peripheral-to-central remapping would have occurred, in my view this should have led to a preservation of the f/mOR bundles in the patients, and so these should have been less distinguishable between patients and controls.

Moreover, the authors do not discuss the possibility that this rather unexpected finding might be an indication for a weakness in their method. For example, the probabilistic atlas that they apply to determine the f/m/p OR bundles may not be tuned towards use in a cohort of mostly elderly people. In fact, the pOR is found in only a relatively small group of participants. Peripheral OR was only found in 2/3 of the participants, compared to > 4/5 for the f/m OR. Neither an explanation nor speculation is provided for this. Although imputation is performed to remedy this, I still find this necessity somewhat disconcerting for their approach. Moreover, some of the f/m/pOR ROC curves contain odd "straight sections" but their origin is not explained. Is this due to the bundles not being found in all participants?

The authors should better explain why the glaucoma trained CNN was applied to an AMD population. In particular, regarding the AMD group, one of the inclusion criteria for the study participants appears to be a final logMAR visual acuity of less than or equal to 0.3. This implies that the AMD group may have been biased towards mild cases, which may not yet show degeneration (even though we know more severe cases clearly do). Obviously, in turn, this may limit the glaucoma CNN from finding any effects. Moreover, why not also apply it to an Alzheimer's Disease cohort, as this disease does show some clinical similarities to glaucoma? In fact, could a comorbidity, for example AD, be a cause for the current "unique non-linearity" observed?

Minor issues

In my view, overall the why of this experiment is not well elaborated upon and the aim is poorly formulated. While current studies do indeed have a small sample size, the primary concern of age creating a bias in the results is not highly valid as many studies have performed case-control (age-sex matched) experiments. Nevertheless, the present study uses a thorough approach for creating a well-matched sample. However, in my view showing the graphs that illustrate the matching in the results section is unnecessary. The statistical matching procedure should be described in the methods section, and figure 1 can be put in supplementary materials.

A large number of abbreviations are introduced without explaining them (e.g. TDI, CST, UNC). Presumably, the manuscript was originally written with the method section following the introduction. It looks like the methods section has simply been shifted to the back of the manuscript, without checking how this would affect the readability of the manuscript as a whole. Also many figure captions are minimalistic in their content, making it hard to grasp what exactly is shown in the figure. The supplementary materials completely lack figure titles and captions.

There is an odd tendency to often change the tense in the methods section.

The authors claim that their classification is accurate, but in my view, even the best ROC curves do not imply a high level of accuracy. So, I would suggest refraining from using this term. It is not clear which feature type(s) resulted in this classification, is it a combination of all three of these or just a single tissue type?

Line 152: Authors state that "In dataset A, the glaucoma subjects have lower FA and MK and higher MD than the control subjects (Figure 2). However, while the mean tract profiles differ, the distributions are highly overlapping. Additionally, there are different mean tissue properties in the UNC and CST. Differences in mean tract profiles are small and not localized to the visual system." I would suggest backing up any of these statements with statistics. By just eyeballing the data, differences are not obvious.

The features analyzed are in all cases DTI/fiber properties. The authors themselves state regarding their use of the kurtosis imaging model: (line 100) "Statistics derived from this model are sensitive to biological changes, such as aging and disease, and, when they are used in concert, can help constrain the interpretations of the underlying biological processes". Yet, in the end there is no attempt to interpret the underlying biological processes. Moreover, there is no discussion on the use of FA and MD. Given the crossing fiber problem, in particular, the FA should be interpreted with caution and might not be the proper feature to base a classification on.

Suggestions

Perhaps give the reader a little guidance on what DTI exactly assesses and how? And what the different tissue properties describe (FA/MD)?

I would be very curious to learn how a CNN trained on the entire ORs would perform
And why not also assess the OT?

Using the CNN network is in my point of view one of the most innovative aspects of the study and might even have future clinical implications (e.g. in screening?) but the authors don't touch upon that possibility in the discussion.

Line 163 The OR bundles in red tend to have better performance than the control bundles.
—> I suggest rephrasing, as the bundles themselves don't perform anything.

Line 370: From the train set, 20% was set aside as a validation set for hyperparameter selection.
Please clarify: —> So, 1/5th of the training set, or 20% of the original dataset (train 60%, validate 20%, test 20%)?

Line 216: Higher SHAP values indicate that a feature contributes towards predicting that the subject either has glaucoma or is older. I suggest also explaining how to interpret negative values.

Line 280: Authors suggest that the features of the OR that are correlated with glaucoma are different from those that are correlated with the normal process of aging. Explain what these features are?

Line 455: It would be helpful to provide the layers of 1D convolutional residual networks used.
Furthermore, to evaluate their network performance, I suggest that authors also report the True Negative and False Positive rates and compute F1-scores.

Although their application of Deep Learning is new, it has previously been applied to glaucoma classification based e.g. on retinal fundus images (<https://doi.org/10.1007/s12652-021-02928-0>).
The authors could consider comparing their results to such literature.

Typos

Sentence on line 107-108 is repeated on lines 108-109

Line 280 Sentence reads odd (see * *): “suggests that the features of the OR that are *correlated with affected in glaucoma* are different from those that are correlated with the normal process of aging.”

Line 334 states “Central 14 deg of eccentricity of the visual field. I presume they want to either say “Central 7 deg of eccentricity” or the “Central 14 deg of the visual field” as anything beyond 7 deg was classified as peripheral.

Reviewer #2 (Remarks to the Author):

This study reports the results of a careful analysis of the effects of glaucoma on diffusion-weighted MRI of fiber tracks representing the foveal, macular and peripheral retinal projection zones. MRI data from self-reported cases of glaucoma were extracted from an online multishell DWI ($b=0, 1000, 2000$) database (UK Biobank) and compared to those of healthy controls after matching for age, gender and demographic factors (Fig. 1). Fibers, tracked from LGN to V1 were segmented into 80 bins along the anterior-to-posterior projection, indicate significant differences in fractional anisotropy (FA), mean diffusion (MD) and mean kurtosis (MK) along select segments of the occipital radiation (OR) for foveal and macular, but not for peripheral, fibers (Fig. 2). Control fibre tracks (CST, UNC) showed no differences between groups. A convolutional neural network was fitted to these data, which yielded receiver-operator characteristics (ROC) with areas-under-the-curve (AUC) indicative of class membership (glaucoma, healthy control) for the foveal and macular fibres. Control comparisons suggest that the trained CNN surpasses linear regression models and the trained network is specific to the eye disease of glaucoma. Some point of clarification are needed (see below).

Major points requiring clarification:

- 1) The quality of the meta-data associated with the dw-MRI is somewhat questionable. The use of the self-reported presence or absence of a specific eye disease in a lay population will add an unknown amount of noise into the assignment to class membership (glaucoma, no glaucoma). Also within the glaucoma group there will be variations in disease subtype (open-angle, normal pressure, etc.), along with disease severity, duration and received treatments. The presence of additional common eye disorders (e.g., high myopia, cataract) in these persons will add to the variance. The same would hold for the healthy controls, i.e., those who did not report the presence of glaucoma, but who well could suffer from other diseases that could affect the retina and the visual pathways (e.g., multiple sclerosis, diabetes). Although the authors touch on this topic, a more thorough discussion of the drawbacks of this retrospective approach is needed.
- 2) It is encouraging (if not so surprising) that the trained CNN provides fits to the diffusion parameters of the central retina projections in glaucoma, but not for patients who self-report macular degeneration (MD). Although the same fibre tracks may be affected in both diseases, the CNNs appear to pick up important differences that do not generalise to another (slightly less common) retinal disease (Fig. 5). What happens in the reversed case: apply CNN trained on AMD data to predict glaucoma cases? It should be straightforward to conduct this control analysis to determine the extent of common variance among these two common retinal diseases.
- 3) The SHapley Additive exPlanations (SHAP) value might not be that obvious to the reader, so more information would be required to appreciate the results presented in Fig. 6). The definition of the abscissa also needs further explanation. The final sentence in the legend to Fig. 6, unfortunately, does not help to clarify the issue much. More information is required here.

Minor points requiring clarification:

- 4) l. 107: The sentence beginning with "One of the hypotheses ..." is repeated. Please remove the second usage.
- 5) l. 134 and Suppl. Fig. 1): Were other risk factors for retinal diseases (history of smoking, diabetes, high blood pressure, elevated intraocular pressure, genetics) recorded in the self-reported metadata. If so, how did these factors weigh into the results? If not, mention should be made in the limitations to this study.
- 6) l. 180: How many patients and controls were included in this analysis (Suppl. Fig. 3)?
- 7) l. 236 ff: The sentence beginning with "SHAP values use ..." does not help to clarify much. Please

be more specific here. What exactly is the reader looking at in this figure and why is it important.

8) l. 439 ff: How accurate is the match between retinographically defined projection zones (à la Benson and Winawer) and the AICHA atlas?

9) l. 461 and Suppl. Fig. 2): The sentence beginning with "In subjects where some bundles ..." needs further clarification. Why are there so many missing data for the fibres of the optic radiation but none for the CST and UNC control tracks? Does it have to do with the curvature of the OR? Were Meyer's and Baum's loops (representing upper and lower visual fields, respectively) analysed separately? Some clarification is needed here.

We would like to thank the Reviewers for reading our manuscript and for their many helpful comments. In our comments below, we have responded to these comments point-by-point. We leave the Reviewer comments in black typeface and add our responses in **red typeface**. When referring to line numbers where changes were introduced, we refer to the marked up version of the article, in which all changes have been tracked.

Reviewer #1 (Remarks to the Author):

The authors present a study on the effect of glaucoma on the brain. They take a novel approach by training ML algorithms to classify the data into glaucoma patients and controls. Their main finding is that whereas a CNN can learn to do this, a conventional linear regression approach cannot. Moreover, they find that the ability to do this classification is specific to the OR and fails on other, non-visual, tracts, as well as on classifying age-defined cohorts. This suggests that glaucoma has a unique non-linear influence on the integrity of the OR.

This manuscript is interesting for a few reasons: first, by using data from the UKBiobank, the study analyzes a very large sample size of glaucoma patients. Secondly, the authors use a thorough approach to find closely matching control participants for the glaucoma participants. Thirdly, the authors compare their OR results to those of two non-visual tracts. Fourth, they train a series of CNNs and do some interesting cross comparisons to determine how specific the glaucoma-related findings are.

Thank you for this positive evaluation of our work, and for the summary of the important points in the manuscript.

The manuscript/study also has some weaknesses. Foremost, the separation of the OR into various bundles results in a rather unexpected finding. Namely that the foveal and macular bundles, rather than the peripheral one, allows the classification. This is quite counterintuitive, as glaucoma typically affects the peripheral visual field. Rather than that this unexpected finding results in the authors questioning their approach, they come up with an, in my view, somewhat flawed explanation for it in terms of functional remapping from the peripheral visual field to the center. Secondly, they do not put much effort into attempting to explain what the “unique non-linear influence” might constitute, in particular at the biological level. Thirdly, the present manuscript is not well written and structured, but this could presumably be remedied.

Thank you also for the constructive criticism. We agree with many of these points, and we appreciate the opportunity to revise our article extensively in light of these comments. We will address each of these as they come up in subsequent comments.

Major issues

The authors separate the OR into three bundles (fovea, macular, peripheral). Surprisingly, the foveal and macular OR bundles enable the best classification even though glaucoma is known to foremost affect the peripheral visual field. The authors come up with an explanation for this in terms of plasticity, but I believe their reasoning on this is flawed. In my reasoning, if indeed peripheral-to-central remapping would have occurred, in my view this should have led to a preservation of the f/mOR bundles in the patients, and so these should have been less distinguishable between patients and controls. Moreover, the authors do not discuss the possibility that this rather unexpected finding might be an indication for a weakness in their method. For example, the probabilistic atlas that they apply to determine the f/m/p OR bundles may not be tuned towards use in a cohort of mostly elderly people. In fact, the pOR is found in only a relatively small group of participants. Peripheral OR was only found in 2/3 of the participants, compared to $> 4/5$ for the f/m OR. Neither an explanation nor speculation is provided for this. Although imputation is performed to remedy this, I still find this necessity somewhat disconcerting for their approach. Moreover, some of the f/m/pOR ROC curves contain odd "straight sections" but their origin is not explained. Is this due to the bundles not being found in all participants?

We agree that there are many limitations in our division of the OR into subbundles. There were many subjects where not all sub-bundles were found, so imputation was necessary. The imputation is what causes the odd straight sections in the ROC curves. Additionally, it reduced the streamline count of the sub-bundles that were found to small numbers of streamlines, decreasing the reliability of the results. The peripheral OR was affected the most by these two phenomena. Thus, we agree that a likely explanation for the differences between the peripheral OR and the macular/foveal OR was our methodology. However, this subdivision is also not material for our main conclusions. Therefore, and in light of these limitations,, we decided to remove the division into sub-bundles from the paper, and simply compare the entire OR to the other control bundles. This reduces the number of missing bundles dramatically (only ~5% of subjects have a missing left OR, for example. As the Reviewer pointed out, ~33% had a missing peripheral OR and ~15% had a missing macular OR, so this is a substantial improvement). It also makes the paper easier to understand, by focusing the paper on the difference between OR and controls, which is our main result.

The authors should better explain why the glaucoma trained CNN was applied to an AMD population. In particular, regarding the AMD group, one of the inclusion criteria for the study participants appears to be a final logMAR visual acuity of less than or equal to 0.3. This implies that the AMD group may have been biased towards mild cases, which may not yet show degeneration (even though we know more severe cases clearly do). Obviously, in turn, this may limit the glaucoma CNN from finding any effects.

We applied the glaucoma trained CNN to the AMD population to address the question of specificity of the OR effects. If the neural network was picking up changes that arise from any significant change to the visual input, the network would presumably also correctly classify

participants with AMD. However, this is not what we found. Instead, the network seems to detect glaucoma-specific characteristics of OR tissue properties. We've added several sentences in the Introduction (L73-L83) to explain this. Furthermore, to alleviate concerns regarding the use of only subjects with relatively intact visual acuity, we re-ran the analyses without this exclusion criterion, which increases the participant pool from N=81 to N=93, and found that the results do not change. We now note this in the Methods section L299: "We re-ran all analyses..."

Moreover, why not also apply it to an Alzheimer's Disease cohort, as this disease does show some clinical similarities to glaucoma? In fact, could a comorbidity, for example AD, be a cause for the current "unique non-linearity" observed?

The relationship between Alzheimer's Disease and glaucoma is indeed of significant interest. However, the UK Biobank sample that we studied does not provide sufficient data to study this relationship using the paradigm that we pursued in this paper. This is because there are only very few participants that have been classified as having AD and also have a dMRI measurement. Nevertheless, the idea that AD and glaucoma may be intertwined is quite compelling and we now mention this idea in the Discussion, with suggestions for future research in this direction. We also mention the possibility of a hidden co-morbidity.

Minor issues

In my view, overall the why of this experiment is not well elaborated upon and the aim is poorly formulated. While current studies do indeed have a small sample size, the primary concern of age creating a bias in the results is not highly valid as many studies have performed case-control (age-sex matched) experiments. Nevertheless, the present study uses a thorough approach for creating a well-matched sample. However, in my view showing the graphs that illustrate the matching in the results section is unnecessary. The statistical matching procedure should be described in the methods section, and figure 1 can be put in supplementary materials.

We appreciate the opportunity to sharpen the importance of a large sample size for the study undertaken. We believe that the large sample size is crucially important for a study that uses non-linear machine learning methods. The sample size used here is approximately *an order of magnitude* larger than any previous study, providing unprecedented opportunities to study the kinds of effects that are documented here. However, we acknowledge that this was not well-described in our original submission. We have accordingly refocused that section of the Introduction on the advantages of large datasets for machine learning methods. As suggested, we also moved Figure 1 to the supplementary materials and the more detailed description of matching to the Methods.

A large number of abbreviations are introduced without explaining them (e.g. TDI, CST, UNC). Presumably, the manuscript was originally written with the method section following the introduction. It looks like the methods section has simply been shifted to the back of the

manuscript, without checking how this would affect the readability of the manuscript as a whole. Also many figure captions are minimalistic in their content, making it hard to grasp what exactly is shown in the figure. The supplementary materials completely lack figure titles and captions.

This has been corrected. All abbreviations have been explained at the first instance in the manuscript and we have improved the figure titles and captions. Thank you for this comment, which has improved the read ability of the manuscript.

There is an odd tendency to often change the tense in the methods section.

This has been corrected.

The authors claim that their classification is accurate, but in my view, even the best ROC curves do not imply a high level of accuracy. So, I would suggest refraining from using this term.

That is correct, we removed all references to accuracy from the paper, or used relative accuracy (ie, the CNN OR is more accurate than the CNNs trained on control bundles).

It is not clear which feature type(s) resulted in this classification, is it a combination of all three of these or just a single tissue type?

This is explained in the Methods briefly, but we have now also added it into the Results section as well (L131 - L133). All three tissue properties from both hemispheres are used for classification.

Line 152: Authors state that "In dataset A, the glaucoma subjects have lower FA and MK and higher MD than the control subjects (Figure 2). However, while the mean tract profiles differ, the distributions are highly overlapping. Additionally, there are different mean tissue properties in the UNC and CST. Differences in mean tract profiles are small and not localized to the visual system."

I would suggest backing up any of these statements with statistics. By just eyeballing the data, differences are not obvious.

Important information about the figure was missing from the caption. There are lines showing the 95% confidence interval as well as the interquartile range. This is now added to the caption and specifically referenced in the text.

The features analyzed are in all cases DTI/fiber properties. The authors themselves state regarding their use of the kurtosis imaging model: (line 100) "Statistics derived from this model are sensitive to biological changes, such as aging and disease, and, when they are used in concert, can help constrain the interpretations of the underlying biological processes". Yet, in the end there is no attempt to interpret the underlying biological processes. Moreover, there is no

discussion on the use of FA and MD. Given the crossing fiber problem, in particular, the FA should be interpreted with caution and might not be the proper feature to base a classification on.

We have added biological interpretation of the tissue properties that arise from the SHAP analysis to the Discussion. We agree that interpretation of dMRI-derived tissue properties needs to be done with caution and we have also added a sentence explaining this in the Discussion (L214-L216)

Suggestions

Perhaps give the reader a little guidance on what DTI exactly assesses and how? And what the different tissue properties describe (FA/MD)?

Good point, this is now added in the Introduction on L66-L70, with the sentence: “The tissue properties we used were...”

I would be very curious to learn how a CNN trained on the entire ORs would perform
And why not also assess the OT?

Due in part to limitations highlighted by the other reviewer, we are now exclusively using the entire ORs for the entire analysis. For the OT, this tract is even harder to recognize reliably and automatically at scale. However, it is a future plan and is now also discussed in the Discussion (L276-279).

Using the CNN network is in my point of view one of the most innovative aspects of the study and might even have future clinical implications (e.g. in screening?) but the authors don't touch upon that possibility in the discussion.

Great point! We have added a paragraph to the Discussion to highlight these possibilities: “Our study contributes...” (L285-293)

Line 163 The OR bundles in red tend to have better performance than the control bundles.
→ I suggest rephrasing, as the bundles themselves don't perform anything.

We removed the word 'performance' from many places and replaced it with more specific words (mostly AUC).

Line 370: From the train set, 20% was set aside as a validation set for hyperparameter selection. Please clarify: → So, 1/5th of the training set, or 20% of the original dataset (train 60%, validate 20%, test 20%)?

We now clarify that it is the former, ie: 64% for the train set, 20% for the test set, and 16% for the validation set.

Line 216: Higher SHAP values indicate that a feature contributes towards predicting that the subject either has glaucoma or is older. I suggest also explaining how to interpret negative values.

This is now also explained, with the magnitude of the SHAP value also explained.

Line 280: Authors suggest that the features of the OR that are correlated with glaucoma are different from those that are correlated with the normal process of aging. Explain what these features are?

In this part of the discussion, we now highlight the different areas in the SHAP plot where SHAPs were different between the age and glaucoma classification CNNs.

Line 455: It would be helpful to provide the layers of 1D convolutional residual networks used. Furthermore, to evaluate their network performance, I suggest that authors also report the True Negative and False Positive rates and compute F1-scores.

Good point, the network architecture is now explained in the machine learning section of the methods section. For network performance, we have chosen to use ROC curves and AUC. Given that our sample is perfectly balanced, the ROC/AUC provides a comprehensive and robust way to evaluate model performance across a range of decision thresholds, not just a single point estimate. It's important to note that True Negative and False Positive rates are indeed incorporated within the ROC curve at specific thresholds. However, the advantage of the ROC curve is that it accounts for any choice of criterion, providing a more complete picture of model performance. We have added a sentence to the first ROC figure caption emphasizing this point.

Although their application of Deep Learning is new, it has previously been applied to glaucoma classification based e.g. on retinal fundus images (<https://doi.org/10.1007/s12652-021-02928-0>). The authors could consider comparing their results to such literature.

Yes, we now compare our application of deep learning to theirs in the discussion, in a new paragraph which starts "Our study contributes..."

Typos

Sentence on line 107-108 is repeated on lines 108-109

Line 280 Sentence reads odd (see * *): "suggests that the features of the OR that are *correlated with affected in glaucoma* are different from those that are correlated with the normal process of aging."

Thank you, corrected.

Line 334 states “Central 14 deg of eccentricity of the visual field. I presume they want to either say “Central 7 deg of eccentricity” or the “Central 14 deg of the visual field” as anything beyond 7 deg was classified as peripheral.

As we no longer use the sub-bundle delineation, this is no longer relevant.

Reviewer #2 (Remarks to the Author):

This study reports the results of a careful analysis of the effects of glaucoma on diffusion-weighted MRI of fiber tracks representing the foveal, macular and peripheral retinal projection zones. MRI data from self-reported cases of glaucoma were extracted from an online multishell DWI ($b=0, 1000, 2000$) database (UK Biobank) and compared to those of healthy controls after matching for age, gender and demographic factors (Fig. 1). Fibers, tracked from LGN to V1 were segmented into 80 bins along the anterior-to-posterior projection, indicate significant differences in fractional anisotropy (FA), mean diffusion (MD) and mean kurtosis (MK) along select segments of the occipital radiation (OR) for foveal and macular, but not for peripheral, fibers (Fig. 2). Control fibre tracks (CST, UNC) showed no differences between groups. A convolutional neural network was fitted to these data, which yielded receiver-operator characteristics (ROC) with areas-under-the-curve (AUC) indicative of class membership (glaucoma, healthy control) for the foveal and macular fibres. Control comparisons suggest that the trained CNN surpasses linear regression models and the trained network is specific to the eye disease of glaucoma. Some point of clarification are needed (see below).

Major points requiring clarification:

1) The quality of the meta-data associated with the dw-MRI is somewhat questionable. The use of the self-reported presence or absence of a specific eye disease in a lay population will add an unknown amount of noise into the assignment to class membership (glaucoma, no glaucoma). Also within the glaucoma group there will be variations in disease subtype (open-angle, normal pressure, etc.), along with disease severity, duration and received treatments. The presence of additional common eye disorders (e.g., high myopia, cataract) in these persons will add to the variance. The same would hold for the healthy controls, i.e., those who did not report the presence of glaucoma, but who well could suffer from other diseases that could affect the retina and the visual pathways (e.g., multiple sclerosis, diabetes). Although the authors touch on this topic, a more thorough discussion of the drawbacks of this retrospective approach is needed.

These are all good points. We extended the Discussion section about the limitations of the prospective approach, to include these issues as well.

2) It is encouraging (if not so surprising) that the trained CNN provides fits to the diffusion parameters of the central retina projections in glaucoma, but not for patients who self-report macular degeneration (MD). Although the same fibre tracks may be affected in both diseases,

the CNNs appear to pick up important differences that do not generalise to another (slightly less common) retinal disease (Fig. 5). What happens in the reversed case: apply CNN trained on AMD data to predict glaucoma cases? It should be straightforward to conduct this control analysis to determine the extent of common variance among these two common retinal diseases.

This would be really interesting to test, in order to establish a double dissociation. Unfortunately, this control analysis is not possible with the current dataset. The reason we only use glaucoma and age trained CNNs is because we can get large enough datasets to train those CNNs. But, there are only 81 AMD subjects in the sample (93 when including those with very poor visual acuity), so these subjects can only be used as test sets for other networks. This is now explained in the Introduction and in the Discussion.

3) The SHapley Additive exPlanations (SHAP) value might not be that obvious to the reader, so more information would be required to appreciate the results presented in Fig. 6). The definition of the abscissa also needs further explanation. The final sentence in the legend to Fig. 6, unfortunately, does not help to clarify the issue much. More information is required here.

We added more information on how SHAP values are calculated, and further explained how to interpret the abscissa.

Minor points requiring clarification:

4) l. 107: The sentence beginning with "One of the hypotheses ..." is repeated. Please remove the second usage.

Corrected.

5) l. 134 and Suppl. Fig. 1): Were other risk factors for retinal diseases (history of smoking, diabetes, high blood pressure, elevated intraocular pressure, genetics) recorded in the self-reported metadata. If so, how did these factors weigh into the results? If not, mention should be made in the limitations to this study.

We discuss limitations such as these in the paragraph "Importantly, the present study is purely correlational..." . The risk factors you mentioned are now mentioned specifically in that paragraph (L253-255). For this study, we did not want to add too many confounders to the statistical matching, as we would then have to accept a less tight match on the confounders we considered to be more important based on previous studies (age, sex, socioeconomic background, cardiovascular health).

6) l. 180: How many patients and controls were included in this analysis (Suppl. Fig. 3)?

How this ROC is made is now better clarified in the caption, including the number of patients and controls used in its test dataset.

7) l. 236 ff: The sentence beginning with "SHAP values use ..." does not help to clarify much. Please be more specific here. What exactly is the reader looking at in this figure and why is it important.

We added more explanation of the SHAP calculation (L154 - L159) and several sentences that are more specific about what differences to look for in the figure and how they are significant (Figure 5 caption)

8) l. 439 ff: How accurate is the match between retinotopically defined projection zones (a la Benson and Winawer) and the AICHA atlas?

We removed the division into sub-bundles, so the Benson and Winawer atlas is no longer used. Before we removed the sub-bundles, we restricted the retinotopically defined projection zones to the AICHA atlas before using them, although they were largely overlapping.

9) l. 461 and Suppl. Fig. 2): The sentence beginning with "In subjects where some bundles ..." needs further clarification. Why are there so many missing data for the fibres of the optic radiation but none for the CST and UNC control tracks? Does it have to do with the curvature of the OR? Were Meyer's and Baum's loops (representing upper and lower visual fields, respectively) analysed separately? Some clarification is needed here.

We were unable to delineate the retinotopically defined sub-bundles of the OR in many subjects, however we have now removed that division into sub-bundles. This significantly reduced the amount of missing data. Still, the OR is more often not found than the controls. This is due to the curvature of the OR and that is now clarified in the machine learning section of the Results (L126-129): "Note that the OR is found less often..." .

Reviewers' comments:

Reviewer #1 (Remarks to the Author):

The authors have improved their manuscript substantially, and addressed a number of major concerns. However, various others remain and some new concerns have now emerged, even concerning doubt about the overall claim of the manuscript. What stands in the paper is their innovative way of testing matters, but I am unconvinced that the authors find strong evidence for their claim that glaucoma may be associated with a unique non-linear signature in OR tissue properties.

Major issues.

In particular, the main concern derives from the, in my view, relatively small, increase in AUC from 0.63 for the logistic regression to 0.69 for the CNN (roughly 10%). Finding an increase compared to logistic regression is not strong enough to claim relevance and drawing far reaching conclusions. Does an increase from a weak AUC of 0.63 to a slightly less weak AUC of 0.69 using computationally much more demanding resources, and potentially much more complex models, warrant this?

In fact, I am not aware of any method that can indicate if the increased performance was worth it relative to the enlarged model complexity. Specifically, if you want to compare a neural network model to a classic model. Irrespective, in terms of interpretability, classical models are preferred over black box processes when accuracy doesn't increase by a substantial amount.

Isn't their result in fact an argument to argue that there is little to no reason for going to such much more complex models? And doesn't it suggest that presently there is no strong reason to assume that glaucoma is associated with a unique non-linear signature in OR tissue properties?

Moreover, I think it's important to discuss the relative impact in terms of false negatives and false positives, as even though the absolute value increased, it does not necessarily improve the classification by that much (a change from 0.98 to 0.99 would have a bigger impact). Having more information on the classification via Confusion matrices could be interesting to add to emphasize the interpretation of the increase in accuracy.

Unless the authors have strong arguments that support their present views, I would suggest removing this claim and discussing findings in a much more balanced way. Ultimately, whether the method is worth it seems quite a subjective, semantic or even philosophical matter that may be hard to objectify. It would be nice if they discuss this e.g. in terms of an open question.

Given the innovative methods used, I still consider the manuscript of value. But this should be enhanced by adding a clear section on the limitations. How can future studies improve on this study? Can the CNN be improved (e.g. U-Net), can the data selection be improved (other inclusion/exclusion criteria), can the metrics be improved (e.g. use of fiber cross-section/fiber density)?

Moreover, given that the comparison of the CNN to the logistic regression is the primary focus of the manuscript, it is at least odd that none of the figures directly compare their AUCs. When doing so informally myself, it is fairly obvious that the curves differ only little. I would suggest merging figures

1 and 2 and use appropriate colors and line texture to distinguish between curves. That would much better tell this manuscript's story.

Another concern relates to the attempt to explain the CNN's performance using the SHAP values. Using SHAP values to describe feature importance is good for explainability of a classification algorithm. However, using such values for explaining the relationship between features and classification results for a model that does not have an acceptable accuracy/performance does not seem very logical to me. When the predictions are only marginally accurate, what is the meaning of finding significance of features for those predictions? How indicative are the SHAP values exactly? What is considered a "high" SHAP value with high indicative power? Now it seems as if they are examining if the SHAP value is positive or negative and drawing conclusions based on that. What is the exact meaning of the numerical values? Substantiate your conclusions and discuss matters, again in a balanced way, given the limited gain obtained with the CNNs..

Minor issues and typos/grammar:

152 "... to interpret the relationship between OR tissue properties and the CNN classifications." Confusing. Do the authors mean "classification performance" or "classification results"?  Do they mean:

"... to interpret the relationship between OR tissue properties on the one hand and CNN classification results on the other."?

15- Convoluted and somewhat confusing sentence: "A convolutional neural network (CNN) classified whether a subject has glaucoma using information from the primary visual connection to cortex (the optic radiations, OR), with significantly higher accuracy than CNNs using information from non-visual brain connections." Since the objective of this sentence is to demonstrate the comparison, consider rephrasing it as "A comparison between convolutional neural networks (CNNs) revealed that those utilizing information from the primary visual connection to the cortex, known as the optic radiations (OR), exhibited higher accuracy in classifying subjects with glaucoma when contrasted with CNNs relying on information from non-visual brain connections."

82 - However, the only AUC that is statistically significant is the AUC from the OR tissue properties (AUC=0.69). Can be more clear  shows a statistically significant difference from chance.

Results section - authors use the term "significant" (for instance, line 135) to describe the AUCs of 0.63 and 0.69, while in general perspective this value is lower or close to acceptable, respectively. I would refrain from the term "significant" to describe these values. Or directly show a significant difference, which now appears to be lacking (I couldn't find it, at least).

The authors appear to use the term "classification(s)" in an inappropriate way when discussing the performance of the model. For instance in the description of Figure 4, "No classifications are significantly different from chance." It should be "Classification performance did not differ significantly from chance ... " I would suggest the authors go through their manuscript and correct similar instances .

393 - The authors have included an explanation on the specification of the CNN network. It would have been more clear if they had provided it in a table or figure form as well.

I suggest having the manuscript checked by a native English editor to improve readability.

Line 35: I would rephrase "simpler models".

Line 72: "in most positions" is quite vague. Be more specific.

Figure 2: "Note that the OR in red has better AUC than the control bundles." Maybe change the color, or specify dark red. A better AUC is oddly phrased. Higher, or larger seems more appropriate.

Line 85: "reliable" classification is a bit optimistic given the still fairly weak AUC of 0.69. It is more reliable than 0.63, but still rather weak.

Line 86: "All bundles were not found in all subjects." Could be phrased better, e.g.  Not all bundles were found in each subject.

Line 95 (and Line 98): "this would indicate that the relationship between glaucoma and the tissue properties is linear and does not require the use of a CNN." Might create confusion between logistic regression and linear regression. Deconfuse.

Line 96: "Again, only the AUC from the CNN trained on the OR (0.63) is significantly different from chance." This AUC is not from the CNN, but from the logistic regression.

Figure 5: The colors are explained twice, this can be described more efficiently.

Line 114-116: "I.e., when a value of a feature increases/decreases does it increase or decrease the probability of classifying a participant having glaucoma, for example." This is not a well composed sentence.  Combination of i.e. and for example can be avoided. Comma between increases/decreases and does. a participant having  a participant as having

Line 187: typo "the these"

OR is inconsistently used in plural or singular form (e.g. Line 19: OR are, Line 88: OR is)

Reviewer #2 (Remarks to the Author):

The revised submission clarifies all of the issues that this reviewer posed. There are just one suggestion for the supplementary information and two minor "wording" issues in the revised text that need clarification or modification.

Suggestions for minor changes in revised submission:

1) l. 111: "The same results are found for the right hemisphere." Could the authors add a figure into the Suppl. Materials that shows the results for the RH (parallel to Fig. 1).

2) l. 127: The sentence beginning with "To test whether the results depends on this missingness, ..." is awkward and it should be rephrased to "To test whether the results depends on these differences

in missing data, ...".

3) I. 203: The sentence beginning with "Additionally, low MK indicates that a subject has glaucoma ..." should be rephrased "Additionally, low MK indicates that a subject is more likely to have glaucoma ..."

We thank the reviewers for the time they have taken to read the article again and for their additional constructive comments. We have revised the article based on these comments, and provide a point-by-point response below. Where we have indicated line numbers for changes that have been implemented, these refer to the version of the manuscript where revisions are marked up.

Reviewers' comments:

Reviewer #1 (Remarks to the Author):

The authors have improved their manuscript substantially, and addressed a number of major concerns. However, various others remain and some new concerns have now emerged, even concerning doubt about the overall claim of the manuscript. What stands in the paper is their innovative way of testing matters, but I am unconvinced that the authors find strong evidence for their claim that glaucoma may be associated with a unique non-linear signature in OR tissue properties.

We appreciate the additional comments provided here. Largely, we agree with the reviewer's comment that the findings do not strongly indicate that non-linearities are a crucial component required to understand the correlations between glaucoma and visual white matter tissue properties. In line with this comment, we have changed the title of the article to "Effects of glaucoma specific to optic radiation tissue properties" and we have shifted the emphasis of the paper away from over-interpretation of non-linearities as indicative of some important biological properties. See more details in our comments below.

Major issues.

In particular, the main concern derives from the, in my view, relatively small, increase in AUC from 0.63 for the logistic regression to 0.69 for the CNN (roughly 10%). Finding an increase compared to logistic regression is not strong enough to claim relevance and drawing far reaching conclusions. Does an increase from a weak AUC of 0.63 to a slightly less weak AUC of 0.69 using computationally much more demanding resources, and potentially much more complex models, warrant this?

It is correct that we needed to do a better job directly comparing these AUCs. We used the DeLong test to find that these AUCs are significantly different, with a marginal p-value, $p=0.0303$ (L102-104). We recognize that the p-value is proximate to the conventional threshold of 0.05, indicating marginal significance. We also recognize the inherent trade-offs when implementing more computationally intensive and complex models like CNNs. The practical implications of this marginally significant increase in AUC are now more thoroughly contextualized in the Discussion section (L188-L196)

In fact, I am not aware of any method that can indicate if the increased performance was worth it relative to the enlarged model complexity. Specifically, if you want to compare a neural network model to a classic model. Irrespective, in terms of interpretability, classical models are preferred over black box processes when accuracy doesn't increase by a substantial amount.

The choice to increase model complexity with a neural network is indeed application-dependent and must consider data and computational resources. Nevertheless, cross-validation methods that are used in this study are asymptotically equivalent to standard statistical methods used to adjudicate between models of different complexity, such as the Akaike Information Criterion, as shown in M. Stone, Cross-Validatory Choice and Assessment of Statistical Predictions. *J. R. Stat. Soc. Series B Stat. Methodol.* 36, 111–147, (1974). While classical/linear models offer better interpretability, we believe the performance gains in our specific context could justify the complexity, given the statistically significant difference. However, we understand the benefits of both approaches and we have rephrased the Discussion section to talk about the benefits of both approaches (L188-L196).

Isn't their result in fact an argument to argue that there is little to no reason for going to such much more complex models? And doesn't it suggest that presently there is no strong reason to assume that glaucoma is associated with a unique non-linear signature in OR tissue properties?

The difference in the two approaches is statistically significant, however we have added caveats in the discussion and results as the difference between the CNN and logistic regression performance is small and both AUCs are weak (L107-L108, L159-L161). We also updated the abstract to moderate the claims about the better performance of the CNNs relative to the performance of the logistic regression models. In the Discussion (L195-L196), we also provide a comparison to our recent study that found similar small benefits for use of CNNs in the analysis of tract-profile data in a different task

Moreover, I think it's important to discuss the relative impact in terms of false negatives and false positives, as even though the absolute value increased, it does not necessarily improve the classification by that much (a change from 0.98 to 0.99 would have a bigger impact). Having more information on the classification via Confusion matrices could be interesting to add to emphasize the interpretation of the increase in accuracy.

We added the confusion matrix to the supplement and now reference it in the Results section (L105-L106).

Unless the authors have strong arguments that support their present views, I would suggest removing this claim and discussing findings in a much more balanced way. Ultimately, whether the method is worth it seems quite a subjective, semantic or even philosophical matter that may be hard to objectify. It would be nice if they discuss this e.g. in terms of an open question.

As suggested, we discuss the findings in a more balanced way in the Discussion section (L188-L196).

Given the innovative methods used, I still consider the manuscript of value. But this should be enhanced by adding a clear section on the limitations. How can future studies improve on this study? Can the CNN be improved (e.g. U-Net), can the data selection be improved (other inclusion/exclusion criteria), can the metrics be improved (e.g. use of fiber cross-section/fiber density)?

We added a (mostly) new paragraph in the Discussion (L248-L263) alluding to future steps. We mention other potential techniques which may mitigate the problems arising from dMRI limitations. We also mention ways in which the neural network analysis could be improved and how more clinical information could be brought in.

Moreover, given that the comparison of the CNN to the logistic regression is the primary focus of the manuscript, it is at least odd that none of the figures directly compare their AUCs. When doing so informally myself, it is fairly obvious that the curves differ only little. I would suggest merging figures 1 and 2 and use appropriate colors and line texture to distinguish between curves. That would much better tell this manuscript's story.

The main result of this paper, in our opinion, is that white matter tissue properties in the OR have some small predictive power for glaucoma diagnosis, and outperform the models trained on control bundle tissue properties. This is seen in both the logistic regression results and CNN results. We now de-emphasize the difference between the CNN and logistic regression in the discussion and abstract. We also perform a statistical significance test for this comparison. However we prefer to keep the two figures separate in order to emphasize the difference between the OR and controls, and to not have too many lines on one plot (visual clarity).

Another concern relates to the attempt to explain the CNN's performance using the SHAP values. Using SHAP values to describe feature importance is good for explainability of a classification algorithm. However, using such values for explaining the relationship between features and classification results for a model that does not have an acceptable accuracy/performance does not seem very logical to me. When the predictions are only marginally accurate, what is the meaning of finding significance of features for those

predictions? How indicative are the SHAP values exactly? What is considered a “high” SHAP value with high indicative power? Now it seems as if they are examining if the SHAP value is positive or negative and drawing conclusions based on that. What is the exact meaning of the numerical values? Substantiate your conclusions and discuss matters, again in a balanced way, given the limited gain obtained with the CNNs..

We agree that the SHAP analysis on these models, whose predictions are only marginally accurate, is not very useful. We have removed the SHAP analysis from the paper (L43-L48, L67-L69, L118-L145, L170-L176, L179-L183, L377-L381).

Minor issues and typos/grammar:

152 “... to interpret the relationship between OR tissue properties and the CNN classifications.” Confusing. Do the authors mean “classification performance” or “classification results”?  Do they mean:

“... to interpret the relationship between OR tissue properties on the one hand and CNN classification results on the other.”?

This section is now removed.

15- Convoluted and somewhat confusing sentence: “A convolutional neural network (CNN) classified whether a subject has glaucoma using information from the primary visual connection to cortex (the optic radiations, OR), with significantly higher accuracy than CNNs using information from non-visual brain connections.” Since the objective of this sentence is to demonstrate the comparison, consider rephrasing it as “A comparison between convolutional neural networks (CNNs) revealed that those utilizing information from the primary visual connection to the cortex, known as the optic radiations (OR), exhibited higher accuracy in classifying subjects with glaucoma when contrasted with CNNs relying on information from non-visual brain connections.”

Corrected (in abstract).

82 - However, the only AUC that is statistically significant is the AUC from the OR tissue properties (AUC=0.69). Can be more clear  shows a statistically significant difference from chance.

Corrected (L83).

Results section - authors use the term “significant” (for instance, line 135) to describe the AUCs of 0.63 and 0.69, while in general perspective this value is lower or close to acceptable, respectively. I would refrain from the term “significant” to describe these values.

Or directly show a significant difference, which now appears to be lacking (I couldn't find it, at least).

We added a test for this difference and found that it is marginally significant (L103-105). See also our responses to other comments about tempering the interpretation of this significant difference.

The authors appear to use the term "classification(s)" in an inappropriate way when discussing the performance of the model. For instance in the description of Figure 4, "No classifications are significantly different from chance." It should be "Classification performance did not differ significantly from chance ... " I would suggest the authors go through their manuscript and correct similar instances .

Corrected (Figure 4 caption).

393 - The authors have included an explanation on the specification of the CNN network. It would have been more clear if they had provided it in a table or figure form as well.

Our architecture is identical to the one in Fawaz et al., which already has this figure. We prefer to leave this figure out as it is not the main focus of this paper.

I suggest having the manuscript checked by a native english editor to improve readability.

Thank you for this suggestion. The manuscript has now been thoroughly revised by a native English speaker.

Line 35: I would rephrase "simpler models".

Replaced the word "simpler" with "other" (L35)

Line 72: "in most positions" is quite vague. Be more specific.

We are now more specific: "the glaucoma subjects have lower MK than the control subjects in their optic radiations across the bundle. They have lower FA and higher MD in the posterior OR, but slightly higher FA and lower MD in the anterior OR." (L72-L73)

Figure 2: "Note that the OR in red has better AUC than the control bundles." Maybe change the color, or specify dark red. A better AUC is oddly phrased. Higher, or larger seems more appropriate.

"better AUC" is replaced with "higher AUC" in both places where we used that phrase (Figure 2 caption, Figure S5 caption). We now specify dark red (Figure 2 caption).

Line 85: “reliable” classification is a bit optimistic given the still fairly weak AUC of 0.69. It is more reliable than 0.63, but still rather weak.

We rephrased this to “This indicates that differences in tissue properties between participants with glaucoma and controls are specific to the OR, and much weaker or non-existent in the non-visual control bundles.” (L86-L88)

Line 86: “All bundles were not found in all subjects.” Could be phrased better, e.g.  Not all bundles were found in each subject.

Corrected (L89)

Line 95 (and Line 98): “this would indicate that the relationship between glaucoma and the tissue properties is linear and does not require the use of a CNN.” Might create confusion between logistic regression and linear regression. Deconfuse.

Corrected to: “this would indicate that the relationship between glaucoma and the tissue properties can be captured by a linear model and does not require the use of a CNN” (L99)

Line 96: “Again, only the AUC from the CNN trained on the OR (0.63) is significantly different from chance.” This AUC is not from the CNN, but from the logistic regression.

Corrected (L100)

Figure 5: The colors are explained twice, this can be described more efficiently.

This figure is now removed.

Line 114-116: “i.e., when a value of a feature increases/decreases does it increase or decrease the probability of classifying a participants having glaucoma, for example.” This is not a well composed sentence.  Combination of i.e. and for example can be avoided. Comma between increases/decreases and does. a participants having  a participant as having

This figure is now removed.

Line 187: typo “the these”

Corrected (L217).

OR is inconsistently used in plural or singular form (e.g. Line 19: OR are, Line 88: OR is)

This is corrected to plural form in a few places (L91, Figure S5 caption)

Reviewer #2 (Remarks to the Author):

The revised submission clarifies all of the issues that this reviewer posed. There are just one suggestion for the supplementary information and two minor "wording" issues in the revised text that need clarification or modification.

Suggestions for minor changes in revised submission:

1) l. 111: "The same results are found for the right hemisphere." Could the authors add a figure into the Suppl. Materials that shows the results for the RH (parallel to Fig. 1).

Good point, added (L80).

2) l. 127: The sentence beginning with "To test whether the results depends on this missingness, ..." is awkward and it should be rephrased to "To test whether the results depends on these differences in missing data, ...".

Changed (L92-93)

3) l. 203: The sentence beginning with "Additionally, low MK indicates that a subject has glaucoma ..." should be rephrased "Additionally, low MK indicates that a subject is more likely to have glaucoma ..."

This section is now removed.

REVIEWERS' COMMENTS:

Reviewer #1 (Remarks to the Author):

Authors have handled comments satisfactorily.

I do have a suggestion for the title, which now primarily emphasizes the comparison of optic radiation to other tract tissue properties. However, it seems kind of obvious that tracts in the visual system are better at diagnosing glaucoma than tracts outside of the visual system. Whereas the effects specific to glaucoma compared to AMD are more interesting and relevant.

Therefore, as suggestion: Optic radiation tissue properties specific to glaucoma

One aspect that the authors could still address in their discussion is that they currently consider all glaucoma patients irrespective of disease stage. This is most likely caused by the lack of good information on disease stage in the UKBiobank. Still, for an approach like this to gain relevance, being able to stage disease, or examine effects as a function of stage, would seem quite essential to me.

Reviewer #2 (Remarks to the Author):

The authors have followed up on my suggestions and they have made the requisite revisions to the manuscript.

REVIEWERS' COMMENTS:

Reviewer #1 (Remarks to the Author):

Authors have handled comments satisfactorily.

I do have a suggestion for the title, which now primarily emphasizes the comparison of optic radiation to other tract tissue properties. However, it seems kind of obvious that tracts in the visual system are better at diagnosing glaucoma than tracts outside of the visual system. Whereas the effects specific to glaucoma compared to AMD are more interesting and relevant.

Therefore, as suggestion: Optic radiation tissue properties specific to glaucoma

Instead of adopting this Reviewer's suggestion, we have adopted the Editors' suggestion to use the title "Convolutional neural network-based classification of glaucoma using optic radiation tissue properties," which emphasizes the technical innovation of the work.

One aspect that the authors could still address in their discussion is that they currently consider all glaucoma patients irrespective of disease stage. This is most likely caused by the lack of good information on disease stage in the UKBiobank. Still, for an approach like this to gain relevance, being able to stage disease, or examine effects as a function of stage, would seem quite essential to me.

This is a good point and we have addressed this comment with the addition of two new sentences in the Discussion (L192-L194 in the revised manuscript)

Reviewer #2 (Remarks to the Author):

The authors have followed up on my suggestions and they have made the requisite revisions to the manuscript.